# Identification and Genomic Characterization of Parvovirus B19V Genotype 3 Viruses from Cases of Meningoencephalitis in West Bengal, India

Chitra Pattabiraman,[a] Pramada Prasad,[a] Sampada Sudarshan,[a] Anson K. George,[a] Darshan Sreenivas,[a] Risha Rasheed,[a] Ayushman Ghosh,[a*] Ananya Pal,[b§] Shafeeq K. Shahul Hameed,[a] Bhaswati Bandyopadhyay,[c] Anita Desai,[a] Ravi Vasanthapuram[a]

[a]Department of Neurovirology, National Institute of Mental Health and Neurosciences, Bengaluru, India
[b]Department of Microbiology, Calcutta School of Tropical Medicine, Kolkata, India
[c]Virology Unit, Department of Microbiology, Calcutta School of Tropical Medicine, Kolkata, India

**ABSTRACT** Brain infections are a major public health problem in India and other parts of the world, causing both mortality and lifelong disability. Even after a thorough investigation, many cases remain without an etiological diagnosis. Primate erythroparvovirus 1 (B19V) has been identified as a pathogen associated with undiagnosed meningoencephalitis in other settings, including the United Kingdom, France, and Latvia. Here, we reported 13/403 (3.2%) B19V PCR positive cases of meningoencephalitis in West Bengal, India. The positive samples were mostly from children (10/13, 76.92%) and presented as a spectrum consisting of acute encephalitis (7/13), acute meningoencephalitis (3/13), and meningitis (3/13). Of the 13 cases, 8/13 (61.5%) had no known etiology and 5/13 (38.5%) had a previous etiological diagnosis. The cases did not cluster in time or by location, suggesting sporadic occurrence rather than outbreaks. We were able to retrieve the complete B19V genomes from cerebrospinal fluid (CSF) in 12/13 cases. The sequences clustered into genotype 3b with complete genomes from Brazil, Ghana, and France, and partial genomes from India and Kyrgyzstan. This is the first report of B19V in cases of neurological infections from India. It highlights the need to evaluate the causal relationship between B19V with meningoencephalitis in the country. These were also the first complete genomes of genotype 3b from CSF and will be critical in the evaluation of the relationship between genotypes and disease.

**IMPORTANCE** Cases of meningoencephalitis with no known etiology remain a major challenge to clinical management of brain infections across the world. In this study, we detected and characterized the whole-genome of primate erythroparvovirus 1 (B19V) in cases of meningoencephalitis in India. Our work highlighted the association between B19V and brain infections which has been reported in other countries. Our work also emphasized the need to examine the role of B19V in meningoencephalitis, specifically whether it caused or contributed to the disease together with other pathogens in India. Our study provided the first 12 genomes of B19V from cerebrospinal fluid. These genomes will contribute to an understanding of how the virus is changing across different locations and over time.

**KEYWORDS** B19V, India, brain infections, meningoencephalitis, parvovirus, pathogen genomics, public health, virology

Address correspondence to Chitra Pattabiraman, chitra.nimhans@gmail.com, or Ravi Vasanthapuram, virusravi@gmail.com.

*Present address: Ayushman Ghosha, Department of Biotechnology, Brainware University, Barasat, Kolkata, India.

§Present address: Ananya Pal, Institute of Post-Graduate Medical Education and Research and Seth Sukhlal Karnani Memorial Hospital (IPGME&R and SSKM), Kolkata, India.

The authors declare no conflict of interest.

The viruses of the species primate erythroparvovirus 1 (B19V) belong to the genus *Erythroparvovirus*, subfamily Parvovirinae and family Parvoviridae (1). Parvoviridae is a family of small, nonenveloped, DNA viruses with single-stranded linear genomes (4 to 6 kb) (1).

B19V has been divided into three genotypes based on nucleotide diversity (2). The viral VP2 region is conserved among the genotypes and infection from different genotypes is

cross-protective (3). Genotype 3 has additionally been classified as genotype 3a and 3b with the strains V9 (NC_004295) and D91.1 (AY083234) as the respective prototypes (4, 5). All three genotypes of B19V are known to circulate in different parts of the world (6). The presence of genotypes 1 and 3 have been reported from India in different clinical conditions and widespread circulation of the virus is supported by serological evidence (7–11).

B19V is a highly infectious virus, spread by respiratory secretions, via transfusion of infected blood products and vertically from mother to child (12). It is pathogenic to humans, causing *Erythema infectiosum* in children, arthropathy in adults, fetal loss in pregnant women, a transient aplastic crisis in certain populations, persistent infection, and pure red cell aplasia in immunocompromised individuals (12, 13). The virus has also been implicated in other hematological disorders, however, whether the infection causes these conditions, contributes to them, or is coincidental is not clear (12–14). The full spectrum of diseases caused by B19V and its pathogenesis, therefore, remains to be delineated and understood.

Parvoviruses have been widely implicated in neurological diseases, including but not limited to encephalitis, meningitis, stroke, and neuropathy (12, 15). Most of the reports of an association between B19V and neurological disease are from case studies (16–21). However, the detection of B19V DNA in the cerebrospinal fluid (CSF) in cases of meningoencephalitis and encephalopathies of unknown etiology, including studies in the UK (4.3%, 7/162) and Latvia (7/42, 16.67%) underscore the association (22–25). A review of 81 cases of neurological illness associated with B19V infections revealed that most of the cases occurred in children, DNA was detected in the CSF of 85% of the cases, and central nervous system involvement occurred in 2/3 of the patients (26). In the study of meningoencephalitis in children from the UK, brain abnormalities were recorded by imaging during the acute phase and neurological sequelae were noted in survivors (22). Neurological complications of B19V infection in immunocompetent adults have also been documented (27–29). There has however been limited molecular characterization/genomic analysis of the virus from CSF. The only documented sequences from CSF are partial sequences from Brazil (GenBank accession no. JX267259.1, JX559657.1, and JX559663.1). Here, we reported the detection of B19V for the first time in cases of meningoencephalitis in the state of West Bengal in eastern India. We also recovered complete genomes of B19V from CSF and performed phylogenetic analysis to place these sequences in the global and local context.

## RESULTS

Genotype-specific PCR for B19V was performed on stored nucleic acid extracted from 403 CSF specimens from individuals presenting with neurological illness in West Bengal. Of these cases, 331 had no known etiology and 72 had etiologies previously ascribed based on laboratory findings (Table 1). Genotype-specific PCR for B19V detected 13 genotype 2/3 positive samples among the 403 samples tested (3.2%), no genotype 1 B19V was detected in this study. PCR positive samples were predominantly from children 10/13, 76.9% compared to PCR negative samples (177/390, 45.4%) (Table 1). Positive cases were distributed across the districts of Darjeeling, Bardhaman, Bankura, Puruliya, and Hooghly in the state of West Bengal and a single case from Deogarh district in the state of Bihar (Table 1, Text S1). All positive samples were collected between September 2017 to September 2018. Clinically the positive cases presented as acute encephalitis (53.8%), acute meningoencephalitis (23.1%), and meningitis (23.1%) (Table 1). Apart from febrile illness, change in mental status and neck rigidity, seizures, drowsiness, and lethargy were also reported in multiple cases (Text S1). The 13 B19V positive samples comprised 2.47% of all samples from cases of unknown etiology (8/331) and 6.26% of samples in which other microbiological etiologies were described (5/72) (Table 1, Text S1). These included one sample positive for *Streptococcus pneumoniae* by PCR, one sample positive for Scrub typhus based on serum IgM, two samples positive for Japanese encephalitis virus (JE) based on either CSF or serum IgM, and one sample that was multiple positive for IgM antibodies to JE (CSF and serum), West Nile virus, Dengue virus, and *Leptospira* sp. (Table 1, Text S1). Of the 8 B19V only positive cases, 6 recovered and were discharged, 2 left against medical advice. Of the 5 cases with B19V and other agents, 4 recovered and were discharged and 1 patient died (JE positive) (Text S1).

**TABLE 1** Summary of socio-demographic, clinical, and laboratory findings in cases (*n* = 403)

| Description/categories | B19V PCR positive | B19V PCR negative | Total samples tested |
|---|---|---|---|
| No. | 13 (3.2%) | 390 (96.8%) | 403 |
| Sex | | | |
| Female | 7/13 (53.8%) | 152/390 (38.97%) | 159/403 (39.45%) |
| Male | 6/13 (46.2%) | 238/390 (61.03%) | 244/403 (60.55%) |
| Age (in yrs) | | | |
| 0-1 | 2/13 (15.4%) | 22/390 (5.64%) | 24/403 (5.96%) |
| 1-18 | 8/13 (61.5%) | 155/390 (39.74%) | 163/403 (40.45%) |
| >18 | 3/13 (23.1%) | 213/390 (54.62%) | 216/403 (53.6%) |
| Districts | | | |
| Darjeeling | 2/13 (15.4%) | 27/390 (6.92%) | 29/403 (7.2%) |
| Bardhaman | 6/13 (46.2%) | 167/390 (42.82%) | 173/403 (42.93%) |
| Bankura | 2/13 (15.4%) | 69/390 (17.69%) | 71/403 (17.62%) |
| Puruliya | 1/13 (7.7%) | 20/390 (5.13%) | 21/403 (5.21%) |
| Deogarh, Bihar | 1/13 (7.7%) | 0/390 (0%) | 1/403 (0.25%) |
| Hooghly | 1/13 (7.7%) | 9/390 (2.31%) | 10/403 (2.48%) |
| Others[a] | | 98/390 (25.13%) | 98/403 (24.32%) |
| Clinical diagnosis at admission[b] | | | |
| Acute meningoencephalitis syndrome (AMES) | 3/13 (23.1%) | 102/390 (26.15%) | 105/403 (26.05%) |
| Acute encephalitis syndrome (AES)[c] | 7/13 (53.8%) | 183/390 (46.92%) | 190/403 (47.15%) |
| Meningitis | 3/13 (23.1%) | 63/390 (16.15%) | 66/403 (16.38%) |
| Microbiological diagnosis | | | |
| Unknown | 8/13 (61.5%) | 323/390 (82.82%) | 331/403 (82.13%) |
| Japanese encephalitis virus (JE) | 2/13 (15.4%) | 11/390 (2.82%) | 13/403 (3.23%) |
| *Streptococcus pneumoniae* | 1/13 (7.7%) | 2/390 (0.51%) | 3/403 (0.74%) |
| Multiple pathogen positive | 1/13 (7.7%) | 14/390 (3.59%) | 15/403 (3.72%) |
| Scrub Typhus | 1/13 (7.7%) | 16/390 (4.1%) | 17/403 (4.22%) |
| Others[d] | | 24/390 (4.9%) | 24/390 (4.9%) |

[a]Includes other districts and nearby states.
[b]Information was not available for 42/390 (10.77%) and 42/403 (10.42%) of the negative and total samples tested, respectively.
[c]AES includes one case of acute disseminated encephalomyelitis.
[d]Other pathogens include Chikungunya, Dengue viruses, and *Leptospira* sp.

Complete genomes were recovered from 12/13 B19V genotype 2/3 positive cases by amplicon sequencing (Text S1). We obtained an average of 0.09 million sequencing reads per sample with an average idealized coverage of 9308× (Text S1). All 12 genomes clustered into a separate cluster within genotype 3b in the maximum likelihood phylogenetic tree of complete B19V genomes as well as specific analysis of genotype 3 sequences (Fig. 1A and B). Of the 12 genomes, 4 genomes (BDN/18/005, BDN/18/195, BKR/18/202, and BKR/18/160) were identical except for the number of Ns in each sequence, all other genomes have distinguishing single nucleotide polymorphisms (Text S1). Complete genomes clustered with B19V genotype 3 sequences from Germany, Brazil, Ghana, and the prototype genotype 3b sequence from France (AY083234, D91.1) (Fig. 1B). As partial genomes of B19V are available from India and other Asian countries, we performed a phylogenetic analysis of our sequences in the context of these sequences (Fig. 2). Sequences from the study clustered into a separate clade within genotype 3b among sequences from India and Kyrgyzstan (Fig. 2).

## DISCUSSION

Brain infections of unknown etiology are a major challenge in India and other parts of the world (30). Many different pathogens can cause brain infections and many of them are known to cocirculate in India (31). However, B19V, which is associated with undiagnosed meningoencephalitis in other settings, including the UK (22) and Latvia (23), has not been reported from India. The virus is known to circulate in the country, and genotypes 1 and 3, in particular, have been reported from pediatric hematological diseases, solid tumors, and cardiomyopathy in India, whereas limited sampling of blood donors did not detect the presence of viral DNA

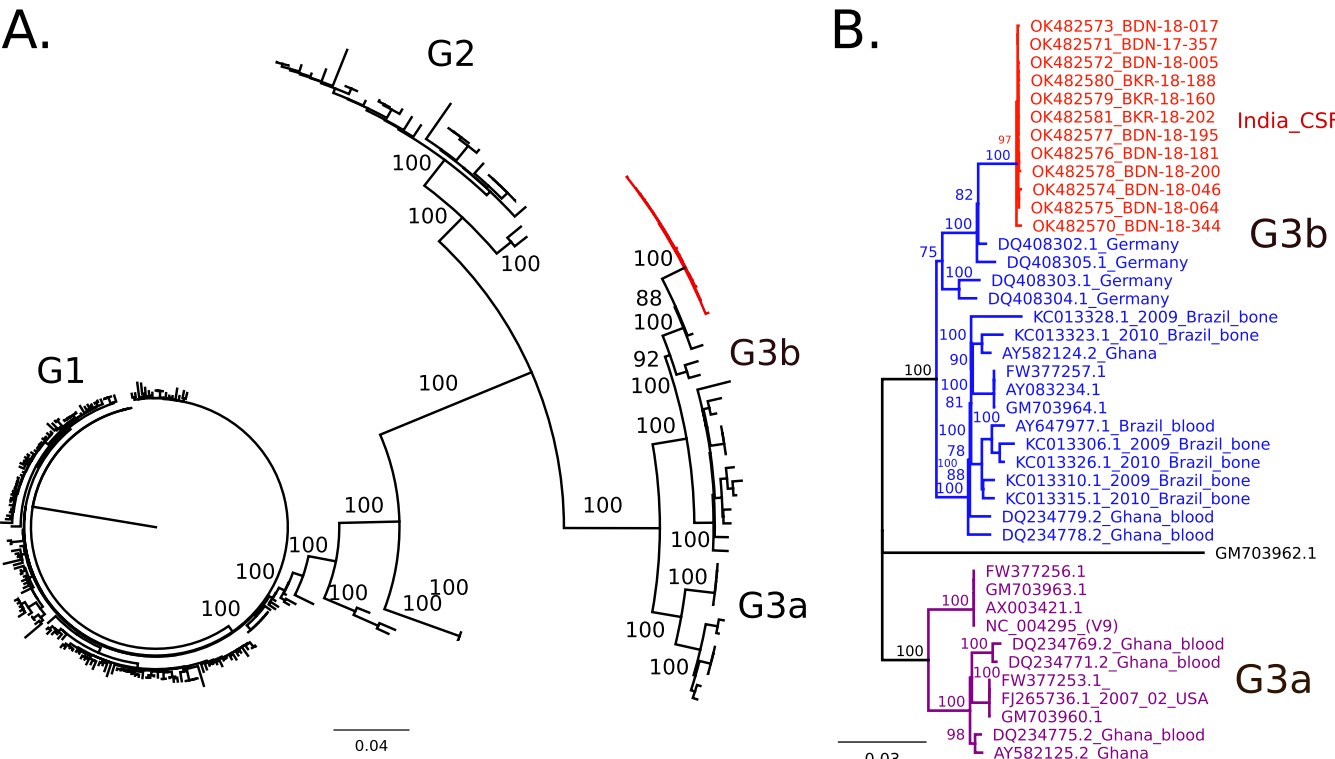

**FIG 1** Phylogenetic analysis of Primate Erythroparvovirus 1 (B19V) complete genomes. Maximum likelihood (ML) trees with nodes with bootstrap values >75 are shown. (A) ML tree of all complete B19V, genomes, including genotypes 1, 2, and 3 (labeled G1, G2, G3, respectively), sequences from this study are highlighted in red. (B) ML tree of genotype 3 sequences with GM703962.1 (G2) as the outgroup. Genotype 3a (purple), genotype 3b (blue), and sequences from this study (red) are included. For sequences from the study, tips are labeled as GenBank ID and sample name. Country and tissue type are indicated. For all other sequences, tips are labeled with GenBank IDs, collection date, country, and tissue type as available.

(6–10). Serological data with seropositivity of 30–50% and a reported increase in positivity with age suggests that the virus is circulating in the population (7, 11, 32).

In this study, we identified B19V DNA in 13 of the 403 (3.2%) samples tested from cases of meningoencephalitis in India (selected to include 331 samples of unknown etiology and

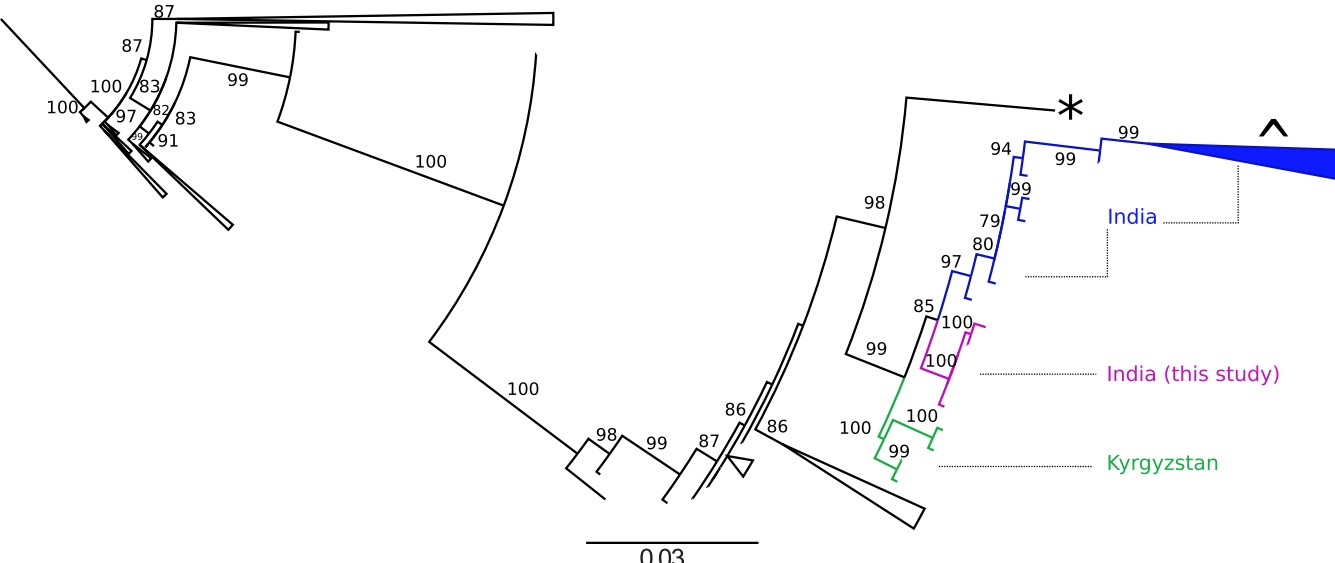

**FIG 2** Phylogenetic analysis of primate erythroparvovirus 1 (B19V) partial genomes from Asia. The maximum likelihood tree of sequences (n = 290) B19V NS1-VP1 region from Asia is shown in the figure. The branch leading up sequences from this study are colored in magenta, sequences from India are in blue (including a collapsed branch [^]) and Kyrgyzstan is in green. Sequences from other branches have been collapsed. Bootstrap values >75 are shown, * represents the prototype sequence NC_004295 (V9, genotype 3a) and NC_000883 (genotype 1) is the outgroup.

72 samples with various known etiologies), from the Eastern state of West Bengal collected between 2016 and 2018. Positive cases were spread throughout the year (September 2017 to September 2018) and across multiple districts suggesting that they were sporadic (Table 1, Text S1). However, local undetected outbreaks cannot be ruled out.

All 13 cases presented either as acute encephalitis, acute meningoencephalitis, or meningitis, characterized by seizures, neck rigidity, drowsiness, and lethargy (Text S1). There was an overrepresentation of children in the positive samples versus the negative samples (Table 1). Most of the positive cases had good immediate outcomes (10/13 recovered and were discharged, 2 left against medical advice), although the possibility of long-term neurological effects needs to be examined. There was 1 reported death among the 13 positive cases, this was in a patient who was also positive for JE by serum and CSF IgM tests (Text S1).

We were able to recover complete genomes from 12/13 positive cases (Table 1). All 12 genomes from this study cluster within a distinct phylogenetic group within genotype 3b (Fig. 1 and 2). The closest complete genomes were from Germany, Ghana, and Brazil and the closest partial genomes were from India and Kyrgyzstan (Fig. 1 and 2) (4, 33, 34). Further genomic characterization of B19V from India and other parts of the world in the context of neurological illness is needed to test for an association of disease with certain genotypes.

One limitation of our study is the lack of supporting serological evidence of B19V infection. Due to the retrospective nature of the study, we only had access to stored nucleic acid and not to the original CSF or serum samples. Additionally, as B19V is a highly infectious virus, we cannot completely rule out the possibility of laboratory contamination. However, the 12 genomes recovered from the study are not identical (Fig. 1B, Text S1) and therefore likely to be from different sources rather than a contamination event. The late Ct value of the sample from which the genome could not be recovered is suggestive of a small amount of virus or sample degradation (Text S1). However, we cannot formally rule out the possibility of a highly divergent B19V which the current primer sets failed to amplify.

B19V has previously been detected by PCR in the blood and CSF of individuals with neurological illnesses (12, 25, 35). Our study was not designed to address a causal link between B19V and brain infections. The detection of B19V DNA in both samples of unknown etiology and those with other known agents raises questions about the direct consequences and causal role of B19V in the context of neurological infections (Table 1). These data must be interpreted in the light of the collective evidence for B19V in brain infections, the observed neurological complications in patients with B19V infections and weighed against the known persistence of B19V in tissues, its presence in asymptomatic individuals, and postmortem brain tissue from individuals with no known illnesses (12, 22, 36, 37). Current evidence supported by cell culture experiments suggests that B19V has a strong tropism for cells of the erythroid lineage (12). Viral DNA has been detected in a wide range of tissues however it is not clear whether the infection in these tissues is productive (12). B19V has been detected in oligodendroglia in different parts of the brain in postmortem tissue (38). These studies suggest that B19V can persist in the brain (12, 37, 38). Early observations suggest a possible role in the demyelination process and support immune-mediated pathophysiology (12, 37, 38).

To our knowledge, this is the first report from India on B19V in meningoencephalitis of unknown etiology. Additionally, even in studies from other settings genomic characterization of B19V from neurological illness is largely absent. Making this the first study to provide 12 sequences of B19V from CSF and place them in their global and local phylogenetic context.

Even as we evaluate its causal role, we argue that B19V is an agent associated with neurological disease in India and that its contribution either alone or in concert with other known causes of brain infections needs to be evaluated further. In the meantime, it should be included in the diagnostic panels for meningoencephalitis of unknown etiology in the country.

## MATERIALS AND METHODS

**Sample collection and storage.** A prospective, 5-year surveillance study for Acute Encephalitis Syndrome was carried out across selected sites in three high burden states of India between 2014 and 2019 (39). All samples were tested for Japanese encephalitis virus (IgM), negative samples were tested for the presence of IgM antibodies to Scrub Typhus, Dengue viruses, and West Nile virus. Additional

molecular tests for *Streptococcus pneumoniae*, *Neisseria meningitidis*, *Haemophilus influenzae*, Herpes simplex virus 1, and Enteroviruses were also performed (39). The study design, testing algorithm, and data collection were described previously (39).

The current study is a retrospective analysis of stored nucleic acid and associated clinical information and laboratory findings from the study described above. Specifically, 403 stored nucleic acid (DNA without RNA removal) from CSF of patients presenting with meningoencephalitis in the state of West Bengal between January 2016 to December 2018 was selected to include 331 samples of unknown etiology and 72 samples with various known etiologies. The use of samples for the study was approved by the Institutional Ethics Committees of Kolkata School of Tropical Medicine, Kolkata, West Bengal, India, and the National Institute of Mental Health and Neurosciences (NIMHANS), Bengaluru, India. Extracted nucleic acid was transported and stored at −80°C until use.

**Genotype-specific PCR for B19V.** Primers and probe sequences for the detection of Parvovirus B19 and its simultaneous classification into genotypes were obtained from the literature (40). B19V primer F-CGCCTGGAACASTGAAACCC, B19V primer R-TCAACCCCWACTAACAGTTC, genotype 1 Probe Cy5-GTTGTAGCTGC ATCGTGGGAAGA-BHQ2, genotypes 2 and 3 Probe-ROX-GTGGTAGCCGCGTCGTGGGAGGA-BHQ2. PCRs were carried out as described previously (40) using ABsolute QPCR Mix (catalog no. AB-1133/A) on an ABI QuantStudioTM 5 real-time PCR system. PCR conditions were as follows: initial denaturation at 95°C for 5 min followed by 50 cycles at 95°C for the 20 s and 50°C for 40 s.

**Amplicon sequencing of B19V.** Based on the qPCR results, multiplex PCR primers spanning the entire B19V genome were designed using genotype 3 V9 (NC_004295) as the reference sequence. These primers would lead to complete recovery of genotype 3 and partial recovery of genotype 2. Details of the primers and protocol for sequencing are provided in the linked protocols.io (41). Briefly, DNA (cDNA and genomic DNA combined) from qPCR positive samples were used as input for a multiplex PCR. The resulting amplicons from each sample were barcoded using the native barcoding kit (NBD104/NBD114, Oxford Nanopore Technologies [ONT]) and sequenced using the ligation kit (SQK-LSK-109, ONT) on the FLO-MIN-106 flow cell on the MinION sequencer (ONT).

**Analysis of sequence data and genome assembly.** The raw sequencing files from the sequencer were base called and demultiplexed using the ONT guppy software (v3.2.10/v5.0.16). Sequencing reads were by 25 bp on both ends and the specific removal of primer sequences was performed using BBDuk (v38.37). Sequences with length >100 bp were assembled to the reference NC_004295 using Minimap2 (v.2.17) (42) within Geneious Prime (2020.0.3). For each sample, a consensus genome was generated with a minimum coverage of 10× for each base and by calling the majority base (most common base) at each position. Consensus sequences were manually examined, edited, and aligned to the reference to ensure the correct reading frame. Annotations were transferred from the reference sequence. Twelve complete genomes were recovered from the study and deposited in GenBank (accession numbers are provided in the data availability section).

**Phylogenetic analysis of B19V.** All available complete and near-complete (4070 to 5596 bp) genome sequences (*n* = 247) of primate erythroparvovirus 1 (B19V) were retrieved from the NCBI database. These represent sequences from across the world (Netherlands 65, Brazil 29, Germany 19, Finland 15, USA 13, Ghana 7, France 6, Serbia 5, China 2, Belgium 2, Vietnam 2, Kenya 1, Bulgaria 1, United Kingdom 1, and data not available 79), where collection dates and release dates are available, release dates were between 1993 and 2021 and collection dates between 2002 and 2017 (Text S1). Multiple sequence alignment of these with the 12 genomes from this study was performed using MUSCLE (v 3.8.425) (43). Alignment was manually inspected and sequences with poor alignment or Ns > 25% were removed. The remaining sequences were realigned and the resulting alignment was used as input for iqtree (v1.6.12) (44) using M13178.1 as the outgroup. A maximum likelihood (ML) based phylogenetic tree with 1000 bootstraps was inferred, using TIM3+F+I+G4 as the best-fit substitution model (of 88 models tested) based on the Bayesian information criterion. The resulting tree was visualized and edited using Figtree (v1.4.4) and nodes with bootstrap values >75 were interpreted. Genotypes were assigned based on metadata available with the sequences. A separate ML tree was constructed for B19V genotype 3 sequences (*n* = 39) with the genotype 2 sequence, GM703962.1, as an outgroup, and TN+F+I+G4 as the best-fit substitution model as described above. To infer the phylogenetic tree for partial sequences from Asia, sequences >300 bp (*n* = 533) were downloaded from NCBI with the associated metadata. These fragments were mapped to the reference and the region with maximum coverage was extracted (1026 bp between the NS1 and VP1 regions). Multiple sequence alignment was performed as described before with B19V references and 12 sequences from this study. Sequences with poor coverage of the region or poor alignment were removed. A phylogenetic tree was constructed with the remaining 290 unique sequences. Of these 290 sequences, the distribution from countries was as follows: Japan (98), China (71), India (43), Kyrgyzstan (23), Iran (17), Israel (13), Kazakhstan (12), Georgia (3), Vietnam (2), Thailand (1), and 5 from this study (7 of the sequences had identical pairs, unique sequences were retained). A maximum-likelihood tree was constructed and interpreted as described above with TIM3e+I+G4 as the best-fit substitution model.

**Data availability.** B19V sequences from the study are available under the following accession numbers: OK482570-OK482581.

## SUPPLEMENTAL MATERIAL

Supplemental material is available online only.
**SUPPLEMENTAL FILE 1**, PDF file, 0.1 MB.

## ACKNOWLEDGMENTS

This work was supported by the DBT/WellcomeTrust India Alliance Fellowship IA/E/ 15/1/502336 awarded to C.P. and core funds of NIMHANS to the Department of Neurovirology.

The funders had no role in study design, data collection, interpretation, or the decision to submit the work for publication.

We declare no conflict of interest.

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
