## [Reviewer comments · Microbiology Spectrum]

Microbiology Spectrum

Identification and genomic characterization of Parvovirus B19V genotype 3 viruses from cases of meningoencephalitis in West Bengal, India

Chitra Pattabiraman, Pramada Prasad, Sampada Sudarshan, Anson George, Darshan Sreenivas, Risha Rasheed, Ayushman Ghosh, Ananya Pal, Shafeeq Hameed, Bhaswati Bandyopadhyay, Anita Desai, and Ravi Vasanthapuram

Corresponding Author(s): Chitra Pattabiraman, National Institute of Mental Health and Neurosciences

Review Timeline:

Submission Date:	November 15, 2021
Editorial Decision:	February 3, 2022
Revision Received:	February 25, 2022
Accepted:	March 19, 2022

Editor: Clinton Jones

Reviewer(s): Disclosure of reviewer identity is with reference to reviewer comments included in decision letter(s). The following individuals involved in review of your submission have agreed to reveal their identity: Saumitra Das (Reviewer #1)

Transaction Report:

DOI: <https://doi.org/10.1128/spectrum.02251-21>

February 3, 2022

Prof. Ravi Vasanthapuram
National Institute of Mental Health and Neurosciences
Department of Neurovirology
Hosur Road
Bengaluru 560029
India

Re: Spectrum02251-21 (**Identification and genomic characterization of Parvovirus B19V genotype 3 viruses from cases of meningoencephalitis in West Bengal, India**)

Dear Prof. Ravi Vasanthapuram:

Thank you for submitting your manuscript to Microbiology Spectrum. As you will see your paper is very close to acceptance. Please modify the manuscript along the lines that the two reviewers have suggested recommended. As these revisions are relatively minor, I expect that you should be able to turn in the revised paper in less than 30 days, if not sooner. However, these changes are important for the manuscript and need to be made. If your manuscript was reviewed, you will find the reviewers' comments below.

When submitting the revised version of your paper, please provide (1) point-by-point responses to the issues I raised in your cover letter, and (2) a PDF file that indicates the changes from the original submission (by highlighting or underlining the changes) as file type "Marked Up Manuscript - For Review Only". Please use this link to submit your revised manuscript. Detailed instructions on submitting your revised paper are below.

Link Not Available

Sincerely,

Clinton Jones

Reviewer comments:

Reviewer #1 (Comments for the Author):

B19V is known to be associated with brain infections in other endemic settings, this work emphasizes the need to examine whether it causes or contributes to the disease together with other pathogens. The authors have attempted to correlate B19V Parvovirus genotype prevalence with the neurological disease, especially among children. Through phylogenetic cluster analyses, the authors have depicted that the cases are sporadic in nature, and not an incidence of any outbreak in India. This study is the first report on association of genotype 3b of B19V Parvovirus with meningoencephalitis in West Bengal, India. In this context, the report is important and significant. Also, the study provides 12 complete genomes of B19V from cerebrospinal fluid, which will be useful information for other studies. Moreover, it encompasses samples from pediatric age group showings signs of acute encephalitis, acute meningoencephalitis and meningitis. Not many studies highlight the clinical manifestations caused by B19V in the endemic settings of India. The manuscript is written well and sounds interesting. However, with only 13 positives samples, it is difficult to establish clear causal correlation.

Minor Comments

1. More number of reference strains around the world (spatial and temporally distributed) would have added strength in the

phylogenetic analyses to reveal a better picture of ancestry of the 12 representative strains included in the study.

2. References should be updated.

Reviewer #3 (Comments for the Author):

The article by Pattabiraman described the identification of parvovirus B19, in samples of nucleic acids, isolated from CSF of patients that presented meningoencephalitis in West Bengal, India. From the 403 samples, only 13 tested positive for parvovirus B19, and in a few cases there was co-infection with other pathogens. The authors recovered full parvovirus B19 genomes and sequence them. The sequences were not identical but group together when compared to full genomes or partial genomes from Asia. This article raises concern about the possibility that parvovirus B19 could be a virus associated to the development of encephalitis, and suggest that future cases include testing for it.

Major points that could be improved are:

-The authors only showed three phylogenetic trees and the supplementary data includes the accession number of the different full genome sequences. As a reader I wonder what are the differences between all the genomes isolated, since the authors mentioned they are not identical. A description of the differences found could be included.

-In the discussion, although it is recognized that parvovirus B19 could only be a bystander agent and it is indicated that its possible role on development of meningitis or other brain inflammations must be explored, it would be nice to see this point with more information if there is some in the literature. Does parvovirus B19 infect cells that are in the brain? Neurons, glia, endothelial cells?

-The references are presented in different formats, this must be corrected, just to mention the two more evident: reference 31 and 39

Several minor points that should be improved.

-Importance section lines 55-56 must be rephrased.

-lines 63 whether it causes...whether it causes

-line 110 Scrub typhus.....Scrub typhus

-line 234 increase is written twice

Preparing Revision Guidelines

- point-by-point responses to the issues I raised in your cover letter
- Upload a compare copy of the manuscript (without figures) as a "Marked-Up Manuscript" file.
- Each figure must be uploaded as a separate file, and any multipanel figures must be assembled into one file.
- Manuscript: A .DOC version of the revised manuscript
- Figures: Editable, high-resolution, individual figure files are required at revision, TIFF or EPS files are preferred

Please return the manuscript within 60 days; if you cannot complete the modification within this time period, please contact me. If you do not wish to modify the manuscript and prefer to submit it to another journal, please notify me of your decision immediately so that the manuscript may be formally withdrawn from consideration by Microbiology Spectrum.

Response to the reviewers

We thank the reviewers for accepting to review our manuscript and for their considered comments. Changes have been made to the manuscript based on these comments and they are addressed in line below -

Reviewer #1 (Comments for the Author):

Minor Comments

1. More number of reference strains around the world (spatial and temporally distributed) would have added strength in the phylogenetic analyses to reveal a better picture of ancestry of the 12 representative strains included in the study.

All complete and near complete genomes of Primate Erythroparvovirus 1 were included for the phylogenetic analysis (n=247). The details of location and collection dates are now provided in Appendix 1 and the methods sections has been modified to reflect this as follows -

All available complete and near complete (4070-5596 bp) genome sequences (n=247) of Primate Erythroparvovirus 1 (B19V) were retrieved from the NCBI database. These represent sequences from across the world (Netherlands 65, Brazil 29, Germany 19, Finland 15, USA 13, Ghana 7, France 6, Serbia 5, China 2, Belgium 2, Vietnam 2, Kenya 1, Bulgaria 1, United Kingdom 1, data NA 79), where collection dates and release dates are available, release dates were between 1993-2021 and collection dates between 2002-2017 (Appendix 1).

2. References should be updated.

References have been reformatted and updated.

Reviewer #3 (Comments for the Author):

Major points that could be improved are:

1. -The authors only showed thre filogenetic tres and the supplementary data includes the accesión number of the different full genomes sequences. As a reader I wonder what are he differences between all the genomes isolated, since the authors mentioned they are not identical. A description of the differences found could be included.

This has been included in Appendix 4 and the following change has been made to the result sections

Of the 12 genomes, 4 genomes (BDN/18/005, BDN/18/195, BKR/18/202 and BKR/18/160) were identical except for the number of Ns in each sequence, all other genomes have distinguishing Single Nucleotide Polymorphisms (Appendix 4).

2. -In the discussion, although it is recognized that parvovirus B19 could only be a bystandars agent and it is inicated that its posible role on developomet of meningitis or other brain inflamations must be explore, it would be nice see this point with more information if there is some in the literature. Does parvovirus B19 infect cell that are in teh brain? Neurons, glia, endotelial cells?

The following paragraph has been added to the Discussion section

Current evidence supported by cell culture experiments suggests that B19V has a strong tropism for cells of the erythroid lineage(12). Viral DNA has been detected in a wide range of tissues however it is not clear whether the infection in these tissues is productive(12). B19V has been detected in oligodendroglia in different parts of the brain in post-mortem tissue(44). These studies suggest that B19V can persist in the brain . Early observations suggest a possible role in the demyelination process and support an immune mediated pathophysiology(12,43,44).

3. -The references are presented in diferente formats, this must be corrected, Just to mention the two more evient: reference 31 and 39

The reference formats have been corrected

Several minor point that should be improved.

4. -Importance section lines 55-56 must be rephrased.

The importance section has been rewritten as follows:

Cases of meningoencephalitis with no known aetiology remain a major challenge to clinical management of brain infections across the world. In this study, we detected and characterized the whole genome of Primate Erythroparvovirus 1 (B19V) in cases of meningoencephalitis in India. Our work highlights the association between B19V and brain infections which has been reported from other countries. Our work also emphasizes the need to examine the role of B19V in meningoencephalitis, specifically, whether it causes or contributes to the disease together with other pathogens in India. Our study provides the first 12 complete genomes of B19V from cerebrospinal fluid. These genomes will contribute to an understanding of how the virus is changing across different locations and over time.

5. -lines 63 wether is causes...wether it causes

corrected

6. -line 110 Scrutb tyfus.....Scrub typhus

This has been corrected

7.-line 234 increase is writen twice

This line has been modified as follows -Serological data - a seropositivity of 30-50% , as well as a reported increase in positivity with age suggests that the virus is circulating in the population (7,11,38).

March 12, 2022

Dr. Chitra Pattabiraman
National Institute of Mental Health and Neurosciences
Department of Neurovirology
Bengaluru 560029
India

Re: Spectrum02251-21R1 (**Identification and genomic characterization of Parvovirus B19V genotype 3 viruses from cases of meningoencephalitis in West Bengal, India**)

Dear Dr. Chitra Pattabiraman:

Your manuscript has been accepted, and I am forwarding it to the ASM Journals Department for publication. You will be notified when your proofs are ready to be viewed.

Sincerely,

Clinton Jones
Editor, Microbiology Spectrum
